# Unbiased Representation of Electronic Health Records for Patient Outcome Prediction

## Abstract

Fairness is one of the newly emerging focuses for building trustworthy artificial intelligence (AI) models. One of the reasons resulting in an unfair model is the algorithm bias towards different groups of samples. A biased model may benefit certain groups but disfavor others. As a result, leaving the fairness problem unresolved might have a significant negative impact, especially in the context of healthcare applications. Integrating both domain-specific and domain-invariant representations, we propose a masked triple attention transformer encoder (MTATE) to learn unbiased and fair data representations of different subpopulations. Specifically, MTATE includes multiple domain classifiers and uses three attention mechanisms to effectively learn the representations of diverse subpopulations. In the experiment on real-world healthcare data, MTATE performed the best among the compared models regarding overall performance and fairness.

## 1 Introduction

Electronic Health Record (EHR) based clinical risk prediction using temporal machine learning (ML) and deep learning (DL) models benefits clinicians for providing precise and timely interventions to high-risk patients and better-allocating hospital resources (Xiao et al., 2018; Shamout et al., 2020). Nevertheless, a long-standing issue that hinders ML and DL model deployment is the concern about model fairness (Gianfrancesco et al., 2018; Ahmad et al., 2020). Fairness in AI/DL refers to a model's ability to make a prediction or decision without any bias against any individual or group (Mehrabi et al., 2021). The behaviors of a biased model often result in two facets: it performs significantly better in certain populations than the others (Parikh et al., 2019), and it makes inequities decisions towards different groups (Panch et al., 2019). Clinical decision-making based upon biased predictions may cause delayed treatment plans for patients in minority groups or misspend healthcare resources where treatment is unnecessary (Gerke et al., 2020).

The data distribution shift problem across different domains is one of the major reasons a model could be biased (Adragna et al., 2020). To address the fairness issue, domain adaptation methods have been developed. The main idea is to learn invariant hidden features across different domains, such that a model would perform similarly no matter to which domain the test cases belong. Pioneer domain adaptation models, including DANN (Ganin et al., 2016), VARADA (Purushotham et al., 2017), and VRNN (Chung et al., 2015), learn invariant hidden features by adding a domain classifier and using a gradient reversal layer to maximize the domain classifier's loss. In return, the learned hidden features are indifferent across domains. Recent work MS-ADS (Khoshnevisan & Chi, 2021) has shown robust performance across minority racial groups by maximizing the distance between the globally-shared presentations with individual local representations of every domain, which effectively consolidates the invariant globally-shared representations across domains. However, it is difficult to align large domain shifts and model complex domain shifts across multiple overlapped domains.

Alternatively, the data distribution shift problem could be addressed using domain-specific bias correction approaches. A recent study showed that features strongly associated with the outcome of interest could be subpopulation-specific (Chouldechova & Roth, 2018). It indicates that lumping together all features from patients with different backgrounds might bury unique domain-specific

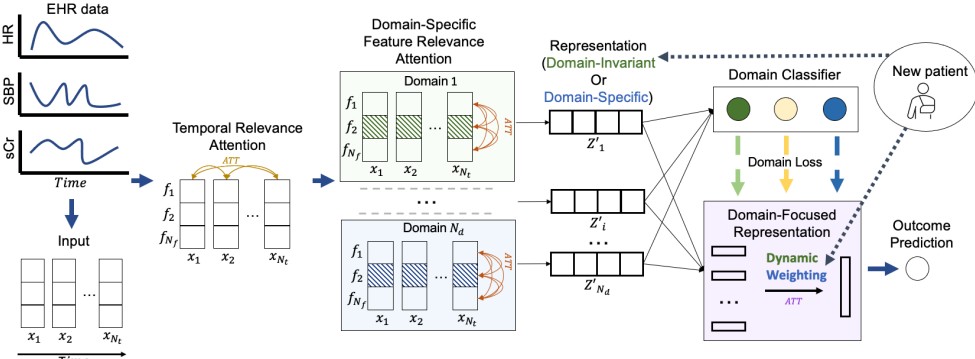

Figure 1: Overall framework of masked triple attention transformer encoder (MTATE). HR, SBP, and sCr stand for heart rate, systolic blood pressure, and serum creatinine, respectively. $x_t$ represents all clinical features at time $t$, $f_i$ represents values of feature $i$ at all time points. $Z'_i$ represents the data representations learned from the $i_{th}$ feature relevance attention module.

characteristics. Afrose et al. proposed to use double prioritized bias correction to train multiple candidate models for different demographic groups (Afrose et al., 2022). Similarly, AC-TPC Lee & Van Der Schaar (2020) and CAMELOT Aguiar et al. (2022) used clustering algorithms to generate representations of patients with similar backgrounds and use cluster-specific representations for the outcome prediction.

In summary, both domain adaptation and domain-specific bias correction approaches address the same fairness issue with different assumptions about the relationships between latent representation and the prediction outcome. The former believes that performance variation across domains would be benefited from invariant feature representation, while the latter affirms domain-specific representations. It remains unclear whether domain-invariant and domain-specific data representation should be used for a prediction task.

To better address the fairness issue, we propose an adaptive multi-task learning algorithm, called MTATE (i.e. Masked Triple Attention Transformer Encoder), to automatically learn and select the optimal and fair data representations instead of explicitly choosing domain adaptation or domain-specific bias correction. Under this setting, both invariant and domain-specific representations are special cases where one of the approaches dominates the data representation. The purpose of MTATE is to generate multiple masked representations of the same data that are attended by both time-wise attention and multiple feature-wise attentions in parallel, where each masked representation corresponds to a specific domain classification task. For example, one of the domain classifiers breaks the patient cohort into subpopulations defined by race, and another classifier is focused on gender. The learned EHR representations could be domain-specific, domain-invariant, or the mix of the two reflected by the domain classification loss values. A low loss value indicates the representation is domain-specific, and a high loss value indicates domain-invariant. The model will compute the representation-wise attention for each individual testing case, leading to personalized data representation for downstream predictive tasks. The overall framework of MTATE is shown in Figure 1. The primary goal of MTATE is to learn an unbiased representation to make fair and precise patient outcome predictions in a real-world healthcare setting.

To demonstrate the effectiveness of MTATE, we focus on rolling mortality predictions for patients with Acute Kidney Injury requiring Dialysis (AKI-D), a severe complication for critically ill patients, with a high in-hospital mortality rate (Lee & Son, 2020). The clinical risk classification for AKI-D patients is challenging due to complex subphenotypes and treatment exposures (Neyra & Nadkarni, 2021; Vaara et al., 2022). There is an urgent need to develop actionable approaches to account for patients' backgrounds and subpopulations for personalized medicine and improve patients-centered outcomes (Chang et al., 2022).

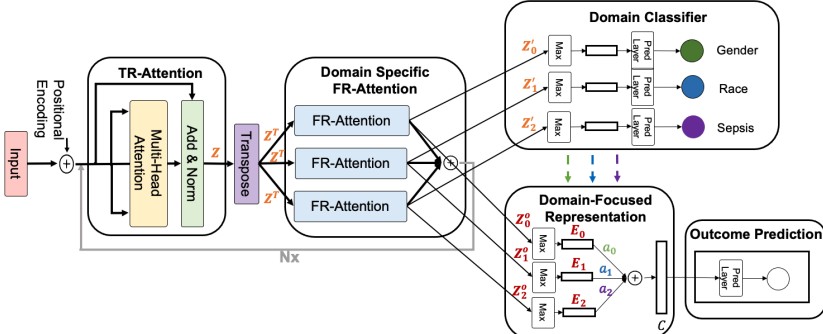

Figure 2: Network structure of the masked triple attention transformer (MTATE) algorithm. The TR-attention and domain specific FR-Attention module can be stacked $N$ times

The contributions of this work are three-fold: 1) To the best of our knowledge, MTATE is the first model that integrates both domain-specific and domain-invariant features in one model. It trains the fair representation and predicts the downstream task altogether, which is critical for reliable clinical outcome predictions for patients with different demographics and clinical backgrounds; 2) MTATE employs time-wise, feature-wise, and representation-wise attention mechanisms to compose data representations for downstream prediction tasks dynamically; and 3) MTATE effectively mitigated the bias towards different subpopulations in the rolling mortality prediction tasks on AKI-D patients and achieved the best performance compared to baselines.

## 2 METHOD

MTATE consists of five components, and the detailed architecture is shown in Figure 2. The first component is a temporal-relevance attention (TR-Attention) module to generate time-wise attention that associates each time step with the other time steps considering all features. The output is a time-attended representation. The second component is a domain-specific feature-relevance attention module to generate feature-wise attentions that associate each feature with the other features considering all the time steps. The outputs are multiple feature-attended representations, one for each domain. The third component is a module consists multiple domain classifiers, and each classifier classifies each feature-attended representation into a predefined domain. The fourth module is a unified data representation module, which uses the representation-wise attention to aggregate feature-attended representations (domain-invariant or domain-specific) to a final representation. The last module is an outcome prediction module where the final representation is used for patient outcome prediction.

### 2.1 NOTATIONS

A patient's EHR data can be represented as $\mathbf{X} = \{\mathbf{x}_1, \mathbf{x}_2, ..., \mathbf{x}_t, ..., \mathbf{x}_{N_t}\}$, $\mathbf{X} \in \mathbb{R}^{N_t \times N_f}$, where $N_f$ is the number of features and $N_t$ is the number of time steps. $\mathbf{x}_t \in \mathbb{R}^{1 \times N_f}$ represents a vector of clinical parameters (e.g., heart rate, blood pressure, etc.) at time step $t$. We consider a binary outcome and domain classification problem in this study. The patient domain class labels are denoted as $\mathbf{dy} \in \mathbb{R}^{N_d}$, where $N_d$ represents the total number of domains, $dy^i \in \{0, 1\}$ represents the label for the $i$-th domain, $1$ and $0$ represent whether a given patient falls in the target domain or not, respectively. The patient outcome label is denoted as $y \in \{0, 1\}$, where $1$ and $0$ represent death and alive before hospital discharge.

### 2.2 TEMPORAL RELEVANCE ATTENTION

The temporal-relevance attention (TR-Attention) module aims to learn the relationships between each time step to other time steps considering all features at each time step. We first use the position

encoding from the original Transformer to encode the relative position information to the input $\mathbf{X}$ and use the multi-head attention mechanism to learn the temporal-relevance attention.

Specifically, query, key, value vectors $(\mathbf{Q}, \mathbf{K}, \mathbf{V})$ are the linear projections of all features at every time steps $\mathbf{X}$. Thus, the attention weights computed from the query and key represent how much focus all features at one single time step is associated with themselves at other time steps. Then, the output of each head $\mathbf{Z^h}$ is the multiplication of value vectors and time-wise attention $\mathbf{A^{TR}}$. The final output of the TR-Attention module $\mathbf{Z} \in \mathbf{R}^{N_t \times N_f}$ is the linear transformation of the concatenation of the output of every head. Lastly, the residual connection and and layer normalization are applied to $\mathbf{Z}$, denoted as $\mathbf{Z} = LayerNorm(\mathbf{Z} + \mathbf{X})$. The temporal-relevance attention of each head is represented as:

$$\mathbf{Q}, \mathbf{K}, \mathbf{V} = \mathbf{X W_Q}, \mathbf{X W_K}, \mathbf{X W_V} \tag{1}$$

$$\mathbf{A^{TR}} = softmax(\frac{QK^T}{\sqrt{d_k}}) \tag{2}$$

$$\mathbf{Z^h} = \mathbf{A^{TR}}V \tag{3}$$

$$\mathbf{Z} = Concat(Z_1^h, ..., Z_i^h, ..., Z_{N_h}^h)W_O \tag{4}$$

For simplicity, we assume all projection matrices $\mathbf{W_Q}, \mathbf{W_K}, \mathbf{W_V}$ have the same dimension $d_k$. Thus, $\mathbf{W_Q}, \mathbf{W_K}, \mathbf{W_V} \in \mathbb{R}^{N_f \times d_k}$, $\mathbf{Q}, \mathbf{K}, \mathbf{V} \in \mathbb{R}^{N_t \times d_k}$, the temporal relevance attention is $\mathbf{A^{TR}} \in \mathbb{R}^{N_t \times N_t}$, the output of each head is $Z^h \in \mathbf{R}^{N_t \times d_k}$. The projection matrix for the final output is $\mathbf{W_O} \in \mathbb{R}^{(N_h d_k) \times N_f}$, where $N_h$ represents the number of heads.

### 2.3 DOMAIN-SPECIFIC FEATURE RELEVANCE ATTENTION

The domain-specific feature relevance attention module aims to learn each domain's diverse and unique latent representation. The module includes a set of parallel sub-modules called feature-relevance attention (FR-Attention), where each FR-Attention module focuses on the representation of one specific domain. Since features for the different domains are not equally important, we randomly masked out some number of latent features along all time steps differently for each FR-Attention module. In return, the masking procedure forces each sub-module to learn different feature focuses for each domain to generate unique domain representations. Then feature-wise attention is computed using the multi-head attention similar to the TR-Attention model. FR-Attention module learns the relationships between each latent feature and the others considering all time steps.

The input of each FR-Attention sub-module is $\mathbf{Z}^T \in \mathbb{R}^{N_f \times N_t}$, which is the transposed output of the TR-Attention module $\mathbf{Z}$. Then, $\mathbf{Z}^T$ is passed through a masking layer, in which $MR \times N_f$ number of latent features are randomly selected and removed from $\mathbf{Z}^T$, where $MR$ represents the masking rate. We denote the masked input of each sub-module as $\mathbf{M} \in \mathbb{R}^{N_l \times N_t}$, where $N_l$ represents the number of features after masking. $\mathbf{M}$ is passed through the multi-head attention block as well as the residual connection and layer normalization. Finally, $\mathbf{M}$ is transposed back to the original form. Then it is passed through a point-wise feed-forward network (FFN) as well as the residual connection and layer normalization to get the final output, denoted as $\mathbf{Z}' \in \mathbf{R}^{N_t \times N_l}$. Please see the detailed architecture and formula for FR-Attention in Appendix Section A.1.

### 2.4 DOMAIN CLASSIFIER

Multiple domain classifiers are used to classify patients into subpopulations based on the latent representation from FR-Attention sub-module. The input of one domain classifier is $\mathbf{Z'_i} \in \mathbf{R}^{N_t \times N_l}$, where $i$ denotes the index of the sub-module or domain. $\mathbf{Z'_i}$ is flattened by taking the max along the time-dimension, then followed by a linear layer with a sigmoid function for binary classification. We use binary cross-entropy for the domain classification loss, denoted as $L_{d_i}$. The domain classifier module assists to learn latent representation for each domain and the domain classification loss is used for generating representation-wise attention in the next module.

## 2.5 DOMAIN-FOCUSED REPRESENTATION AND REPRESENTATION-WISE ATTENTION

While a domain-specific representation module is focused on representing its target domain, the resulting representation from each domain can be domain-specific or domain-invariant according to the domain loss. However, not all representations are equally crucial to outcome prediction. Thus, this module aims to generate the final representation for the outcome prediction considering both domain-specific and domain-invariant representations and their corresponding domain loss. We call this module a domain-focused representation module since both domain-specific and domain-invariant are candidate representations.

The input to this module is a transformed version of the latent representation generated from each FR-Attention module $\mathbf{Z'_i}$. Every $\mathbf{Z'_i}$ gets transformed to its original dimension by adding the masked(removed) feature back with values of 0 so that all latent representations can be aligned in feature space. We denoted this particular form of latent representation as $\mathbf{Z^o_i}$. $\mathbf{Z^o_i}$ is flattened by taking the max along the time dimension, and all flattened vectors are concatenated together, denoted as $\mathbf{E} \in \mathbb{R}^{N_d \times N_f}$. The final representation $\mathbf{C} \in \mathbb{R}^{N_f \times 1}$ is the weighted sum of all the candidate representations, where the weights $\mathbf{a} \in \mathbb{R}^{N_d \times 1}$ are the representation-wise attention (RW-Attention), which is computed based on $\mathbf{E}$ and the domain prediction loss $\mathbf{L_d} \in \mathbb{R}^{N_d \times 1}$ as following:

$$\mathbf{a} = softmax(\tanh(Concat(E, L_d)U_A)W_A) \tag{5}$$

$$C_j = \sum_{i=1}^{N_d} a_i E_{i,j} \tag{6}$$

where $U_A \in \mathbb{R}^{(N_f+1) \times d_a}$ and $W_A \in \mathbb{R}^{d_a \times 1}$ are the projection matrices, $i$ represents the domain index, $j$ represents the feature index.

## 2.6 PATIENT OUTCOME PREDICTION

The final representation $\mathbf{C}$ of EHR is concatenated with all static features, such as demographics and comorbidity, followed by a linear layer with a sigmoid function for the outcome binary prediction. Let the patient outcome label be $y$ and the predicted label be $\hat{y}$, and we use the binary cross entropy as the part of the final loss, denoted as $L_p$. We also constructed the supervised contrastive loss Khosla et al. (2020) to mitigate further the model bias as another part of the final loss. The contrastive loss is denoted as $L_c$. The final prediction loss $L$ is:

$$L = L_p + L_c \tag{7}$$

$$L_p = \sum_{i=1}^{N_s} -(y_i \log(\hat{y_i}) + (1 - y_i) \log(1 - \hat{y_i})) \tag{8}$$

$$L_c = \sum_{j=1}^{N_s} \frac{-1}{N_p} \sum_{p=1}^{N_p} \log \frac{exp(h_j * h_p/\tau)}{\sum_{a=1}^{N_a} exp(h_j * h_a/\tau)} \tag{9}$$

where $N_s, N_p, N_a$ represents the number of all samples, the number of samples having the same labels as the anchor samples $(j)$, and the number of samples having the opposite label to the anchor samples $(j)$. $h$ represents the concatenation of the learned representation $\mathbf{C}$ and the static features. $\tau$ is a scale parameter.

## 3 EXPERIMENTS SETTINGS

In the experiment, we aim to continuously predict AKI-D patients' mortality risk in their dialysis/renal replacement therapy (RRT) duration. More specifically, given a period of EHR in dialysis duration before time $T$, we will continuously predict the mortality risk between $T$ and $T + 72h$.

### 3.1 EXPERIMENT DATA

The study population consists of 570 AKI-D adult patients admitted to ICU at the University Hospital from January 2009 to October 2019. Among them, 237 (41.6 %) died before discharge, and

Table 1: Training, validation and testing data.

|  | Total N Samples (Patient) | Alive N Samples (Patient) | Death in the next 72 hours N Samples (Patient) | Negative to Positive Ratio (Patient) |
|---|---|---|---|---|
| Train | 9979 (432) | 7652 (388) | 2327 (149) | 3:1 (3:1) |
| Valid | 642 (24) | 519 (22) | 123 (9) | 4:1 (2:1) |
| Test | 2712 (114) | 2187 (104) | 525 (31) | 4:1 (3:1) |
| All | 13333 (570) | 10358 (514) | 2975 (189) | 3:1 (3:1) |

333 (58.4%) survived. Patients are excluded if they were diagnosed with end-stage kidney disease (ESKD) before or at the time of hospital admission, are recipients of a kidney transplant, or has RRT less than 72 or greater than 2,000 hours.

Data features include 12 temporal features (systolic blood pressure, diastolic blood pressure, serum creatinine, bicarbonate, hematocrit, potassium, bilirubin, sodium, temperature, white blood cells (WBC) count, heart rate, and respiratory rate) and 11 static features including demographics and comorbidities (age, race, gender, admission weight, body mass index (BMI), Charlson comorbidity score, diabetes, hypertension, cardiovascular disease, Chronic Kidney Disease, and Sepsis). All outliers ($> 97.5\%$ or $< 2.5\%$) were excluded and missing values are imputed with the last observation carried forward (LOCF) method.

To continuously predict mortality risks, we generate 30 samples from each patient's EHR data with random start and end times as long as the duration is greater than 10 time steps. The class label of a sample is whether the patient died (positive) or survived (negative) in the next 72 hours. From 570 AKI-D patients, 13,333 EHR samples are extracted, including 2,975 positive and 10,358 negative samples. As shown in Table 1, all the EHR samples are split into train (75%), validation (5%), and test data (20%) patient-wise, which ensures that samples from the same patients only appear in one of the three sets. Eighteen subpopulations were considered in this study based on nine domains according to patient demographics (i.e., age, gender, race) and commodities (i.e., Charlson score, diabetes, hypertension, cardiovascular disease, chronic kidney disease, and sepsis).

### 3.2 BASELINE METHOD AND FAIRNESS PERFORMANCE METRICS

We compared MTATE with two widely used and accessible sequence DL methods, LSTM and Transformer, one well-known EHR-specific representations method, RETAIN, and one pioneer domain-adaptation method, DANN*. For Transformer, the encoder part of the original Transformer is used. For DANN*, we only use the gradient reversal layer from the original DANN to get domain-invariant representation with all other structures the same as MTATE. For all models, the input data are the EHR temporal features, and static features are concatenated with latent representation before the prediction layer, as described in Section 2.6. We evaluate the performance of all models using traditional performance metrics: Area under the ROC Curve (ROCAUC), Accuracy(ACC), Area under the Precision-Recall Curve (PRAUC) as well as three fairness metrics Demographic Parity Difference (DPD), Equality of Opportunity Difference (EOD) and Equalized Odds Difference (EQOD) (Feldman et al., 2015; Hardt et al., 2016; Afrose et al., 2022) (see fairness metric equations 15, 16, 17 in Appendix). We compare all models on both imbalanced and balanced sets. The positive samples (died) and negative samples (survived) are 1:4 and 1:1, respectively.

## 4 RESULTS AND DISCUSSION

### 4.1 COMPARISON WITH BASELINE METHODS

The overall performance of rolling mortality prediction in the next 72 hours for all test data with an imbalanced positive to negative ratio are shown in Table 2. We also show the performance on the balanced data in Table A1. Regarding the imbalanced test data, Table 2 shows that MTATE outperforms all the compared baseline methods in almost all metrics. LSTM is the most competitive method since it has the same highest ROCAUC as MTATE and the highest PRAUC. Nevertheless, the fairness scores of LSTM are not as good as MTATE. RETAIN, and Transformer have a moderate

Table 2: Performance comparison on mortality prediction in the next 72 hours on imbalanced test data (pos:neg=1:4). DPD, EOD, and EQOD are the lower the better.

| Method | ROCAUC | ACC | PRAUC | DPD | EOD | EQOD |
|---|---|---|---|---|---|---|
| Transformer | 0.71(0.08) | 0.69(0.09) | 0.39(0.17) | 0.18(0.10) | 0.08(0.08) | 0.10(0.05) |
| LSTM | **0.72(0.11)** | 0.77(0.07) | **0.55(0.20)** | 0.12(0.08) | 0.10(0.07) | 0.07(0.04) |
| RETAIN | 0.69(0.12) | 0.78(0.07) | 0.45(0.20) | 0.13(0.12) | 0.10(0.07) | 0.07(0.06) |
| DANN* | 0.60(0.12) | 0.64(0.14) | 0.31(0.16) | 0.23(0.18) | 0.08(0.06) | 0.12(0.08) |
| MTATE (Ours) | **0.72(0.09)** | **0.81(0.07)** | 0.49(0.18) | **0.09(0.07)** | **0.07(0.06)** | **0.05(0.03)** |

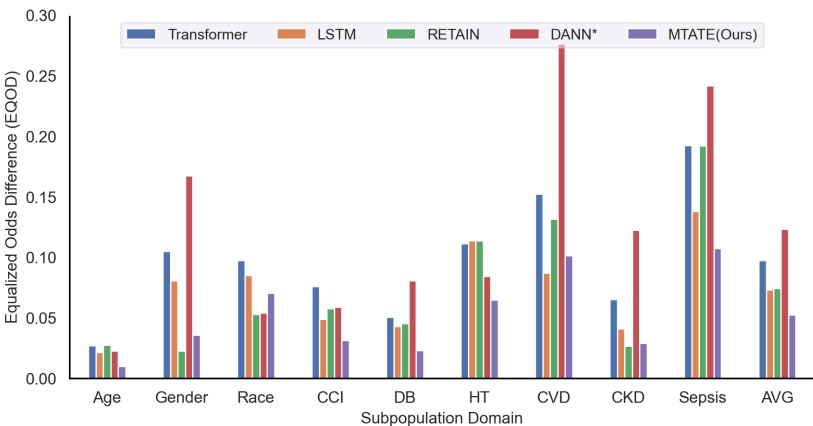

Figure 3: Equalized odds difference for every subpopulation domain. Y-axis represents the EQOD score (the lower the better) and X-axis represents the subpopulation domains, where each domain consists of two subpopulations (e.g., Young ($< 65$ y/o) vs. Old in Age and Sepsis vs. Non-Sepsis). Here, CCI, DB, CVD, and CKD stand for Charlson comorbidity score, diabetes, hypertension, cardiovascular disease, and chronic kidney disease, respectively.

performance. The performance of all the compared algorithms on the balanced test data shows that MTATE has the best ROCAUC, DPD, EOD, and EQOD, and second-best accuracy (see Table A1).

We compare MTATE with all baseline methods within each subpopulation domain. Figure 3 shows that MTATE has the best (lowest) averaged EQOD score. We also compare MTATE with all baselines regarding the difference of PRAUC within each subpopulation domain (e.g., the difference of PRAUC between female and male). The difference in PRAUC for each domain and the averaged score across all domains is in Figure A2. It shows that MTATE has the lowest percentage difference in PRAUC between subpopulations in Age, Gender, and Hypertension domains.

## 4.2 ABLATION STUDY

We conduct an ablation study to test how each component of MTATE performs by removing them from MTATE. Table 3 shows that MTATE has the best performance for almost all metrics except for ROCAUC, which is the second best to MTATE without masking. However, the accuracy of MTATE without masking is $18\%$ lower than MTATE. In addition, RW-ATT is the most effective component since the performance drops the most in the two ablations that are without RW-ATT.

A primary goal of MTATE is to learn fair representations that can be used by a wide spectrum of downstream predictive models. To this end, we test whether the representation learned from MTATE can be used by traditional machine learning methods. The last three lines in Table 3 shows that all three traditional methods, XGboost, SVM, and Random Forest, have achieved similar performance as MTATE and are better than some of the compared deep learning methods. This comparison confirms that MTATE can serve as a pre-trained EHR data representation generator, and the learned representations can be used by downstream prediction tasks implemented with different classifiers.

Table 3: Performance comparison of MTATE and its ablation components for the mortality prediction in the next 72 hours (pos:neg = 1:4). **w/o. DC:** remove all domain classifiers. **w/o. RW-ATT:** remove representation-wise attention. **w/o. DC** & **RW-ATT:** remove both domain classifier and representation-wise attention. **w/o. Masking:** remove masking layers in FR-Attention module. **w/o.** $L_c$ **:** remove contrastive loss.

| Method | ROCAUC | ACC | PRAUC | DPD | EOD | EQOD |
|---|---|---|---|---|---|---|
| w/o. DC | 0.70(0.09) | 0.70(0.07) | 0.47(0.18) | 0.15(0.12) | 0.09(0.07) | 0.09(0.06) |
| w/o. RW-ATT | 0.64(0.13) | 0.72(0.10) | 0.37(0.21) | 0.23(0.20) | 0.12(0.09) | 0.13(0.09) |
| w/o. DC & RW-ATT | 0.63(0.13) | 0.70(0.11) | 0.37(0.21) | 0.25(0.20) | 0.11(0.10) | 0.13(0.09) |
| w/o. Masking | **0.73(0.09)** | 0.63(0.06) | 0.48(0.18) | 0.15(0.12) | 0.09(0.08) | 0.09(0.06) |
| w/o. $L_c$ | 0.71(0.09) | 0.77(0.07) | 0.44(0.17) | 0.11(0.11) | 0.08(0.06) | 0.07(0.05) |
| MTATE | 0.72(0.09) | **0.81(0.07)** | **0.49(0.18)** | **0.09(0.07)** | **0.07(0.06)** | **0.05(0.03)** |
| XGBoost w/ MTATE | 0.69(0.09) | 0.70(0.07) | 0.43(0.17) | 0.09(0.08) | 0.07(0.07) | 0.07(0.03) |
| SVM w/ MTATE | 0.71(0.10) | **0.81(0.06)** | 0.48(0.19) | 0.10(0.08) | 0.07(0.06) | 0.06(0.04) |
| RF w/ MTATE | **0.72(0.09)** | 0.80(0.06) | **0.49(0.17)** | **0.07(0.07)** | **0.06(0.06)** | **0.05(0.03)** |

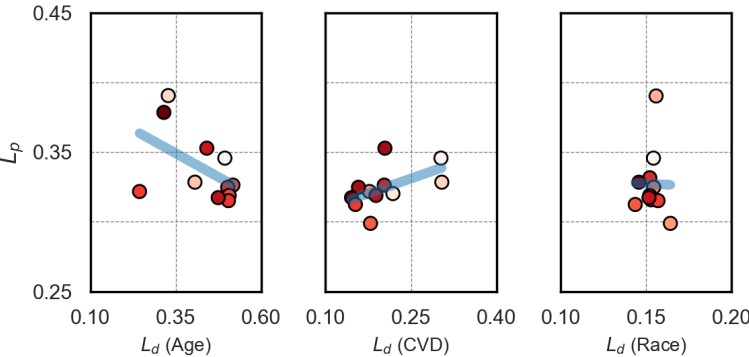

Figure 4: Relationship between outcome loss, domain loss and representation-wise attention. Y-axis represents the outcome loss, X-axis represents the domain loss. The colored dots represent representation-wise attention, and the darker color represents higher attention.

## 4.3 EFFECTIVENESS ASSESSMENT OF RW-ATTENTION

Since RW-Attention is the most effective component in MTATE, we study its relationships with the outcome prediction loss $L_p$ and domain loss $L_d$, as shown in Figure 4. Each dot in the figure presents the averaged value of all samples from the same subpopulation. The figure shows three example domains in two facets. First, the correlation between the outcome prediction loss and the domain loss could be negatively correlated (Age), positive (CVD), or mixed (Race). In the negative correlation scenario, the higher the domain loss, the lower the outcome prediction loss, which suggests the RW-Attention is putting more weight on the representations with larger domain loss (i.e., domain-invariant representations). In the positively correlated scenario, outcome prediction loss decreases with the decrease of domain loss. It suggests the RW-Attention is putting more weight on the representations with smaller domain loss (domain-specific representations). Second, attention weights, as indicated by the color of the dots in the figure, demonstrate the relationship between RW-Attention and the outcome prediction loss. The darker colored dots (greater attention) are almost always related to the lower outcome prediction loss, whether the outcome prediction loss and the domain loss are positively or negatively correlated. This indicates that the RW-Attention module can weigh both domain-specific or domain-invariant representations toward lower outcome prediction loss. Similar patterns in all other domains are in Figure A4.

## 5 CONCLUSION

In this work, we presented MTATE, an attention-based encoder for EHR data. MTATE uses three different attention mechanism (time-relevance, feature-relevance, and representation-wise) to learn unbiased data representations. Experiments on real-world healthcare data demonstrated that MTATE outperforms the compared baseline methods on continuous mortality risk prediction for critically ill AKI-D patients regarding fairness.

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

# A APPENDIX

## A.1 FR-ATTENTION

The FR-Attention sub-module for each head is computed as:

$$\mathbf{Q}', \mathbf{K}', \mathbf{V}' = \mathbf{MU_Q}, \mathbf{MU_K}, \mathbf{MU_V} \tag{10}$$

$$\mathbf{A^{FR}} = softmax(\frac{Q'K'^T}{\sqrt{d'_k}}) \tag{11}$$

$$\mathbf{M'^h} = \mathbf{A^{FR}}V' \tag{12}$$

$$\mathbf{M'} = Concat(M_1'^h, ..., M_i'^h, ..., M_{N_h'}'^h)U_O \tag{13}$$

$$\mathbf{Z'} = \mathbf{max(0, (M')^T W_1 + b_1)W_2 + b_2} \tag{14}$$

Similar to TR-Attention module, we assume all projection matrix $\mathbf{U_Q}, \mathbf{U_K}, \mathbf{U_V}$ have the same dimension $d'_k$. Thus, $\mathbf{U_Q}, \mathbf{U_K}, \mathbf{U_V} \in \mathbb{R}^{N_t \times d'_k}$ and $\mathbf{Q}', \mathbf{K}', \mathbf{V}' \in \mathbb{R}^{N_l \times d'_k}$. The feature-relevance attention $\mathbf{A^{FR}} \in \mathbb{R}^{N_l \times N_l}$. The output of each head is $\mathbf{M'^h} \in \mathbf{R}^{N_l \times d'_k}$. Similar to the TR-Attention module, all outputs from all heads are concatenated to form $\mathbf{M'} \in \mathbf{R}^{N_l \times N_t}$, and linear transformation are applied with the projection matrix $\mathbf{U_O} \in \mathbb{R}^{(N_h' d'_k) \times N_t}$, where $N_h'$ represents the number of head.

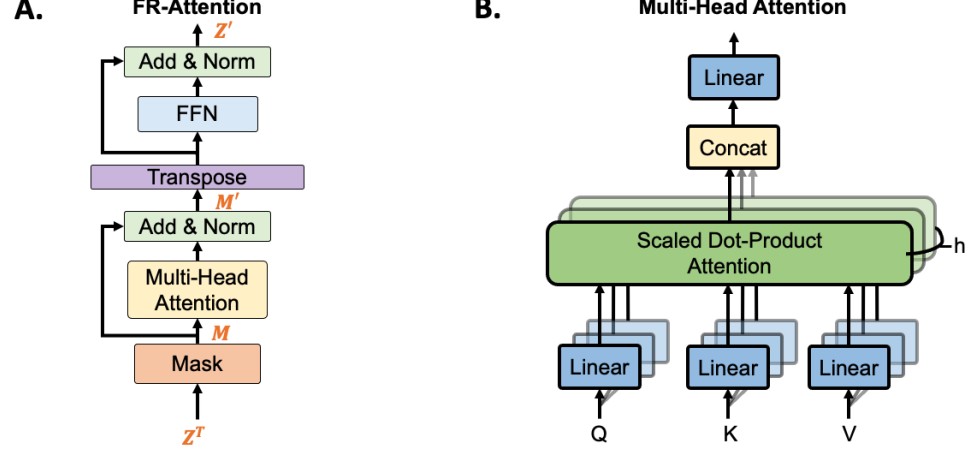

Figure A1: A. The structure of FR-Attention module in MTATE. B. The multi-head attention module from the original Transformer used in MTATE.

## A.2 FAIRNESS METRICS

Demographic parity suggests that a predictor is unbiased if the prediction is independent of the protected attribute (e.g., Age, Gender, and etc.). We denote protected attribute as $A \in a, b$, and $A$ only take two groups $a, b$ (e.g., Young vs Old for Age) for simplicity. Thus, the Demographic parity difference (DPD) is the difference between the two group $a$ and $b$ The formula for Demographic parity difference (DPD) is shown in below:

$$DPD = P(\hat{y} = 1 | A = a) - P(\hat{y} = 1 | A = b) \tag{15}$$

Equality of opportunity suggests that a predictor is unbiased if the true-positive rate between two groups are equal. Similarly, the Equality of opportunity difference (EOD) is the difference between the two group $a$ and $b$. The formula for EOD is shown in below:

$$EOD = P(\hat{y} = 1 | y = 1, A = a) - P(\hat{y} = 1 | y = 1, A = b) \tag{16}$$

Equalized odds suggests that a predictor is unbiased if both the true-positive rate (TPR) and false-positive rate (FPR) between two groups are equal. We computes the Equalized odds Difference (EQOD) as the average of the difference in both TPR and FPR. The formula for EQOD is shown in below:

$$EQOD = (TPR_D + FPR_D)/2 \tag{17}$$
$$TPR_D = P(\hat{y} = 1|y = 1, A = a) - P(\hat{y} = 1|y = 1, A = b) \tag{18}$$
$$FPR_D = P(\hat{y} = 1|y = 0, A = a) - P(\hat{y} = 1|y = 0, A = b) \tag{19}$$

### A.3 PERFORMANCE COMPARISONS

#### A.3.1 OVERALL PERFORMANCE

The following table shows the performance comparison between MTATE and baseline methods for the balanced test data.

Table A1: Balanced performance of MTATE and compared algorithms for the mortality prediction in the next 72 hours (pos:neg = 1:1).

| Method | ROCAUC | ACC | PRAUC | DPD | EOD | EQOD |
|---|---|---|---|---|---|---|
| Transformer | 0.70(0.09) | **0.67(0.09)** | 0.69(0.11) | 0.17(0.13) | 0.15(0.12) | 0.10(0.06) |
| LSTM | 0.71(0.11) | **0.67(0.11)** | **0.76(0.12)** | 0.19(0.12) | 0.20(0.12) | 0.12(0.05) |
| RETAIN | 0.68(0.12) | **0.67(0.11)** | 0.72(0.13) | 0.22(0.13) | 0.21(0.12) | 0.12(0.06) |
| DANN* | 0.58(0.12) | 0.54(0.14) | 0.59(0.12) | 0.21(0.19) | 0.16(0.13) | 0.12(0.08) |
| MTATE (Ours) | **0.73(0.09)** | 0.65(0.11) | 0.75(0.11) | **0.15(0.10)** | **0.14(0.11)** | **0.08(0.05)** |

#### A.3.2 FAIRNESS PERFORMANCE WITHIN EACH DOMAIN

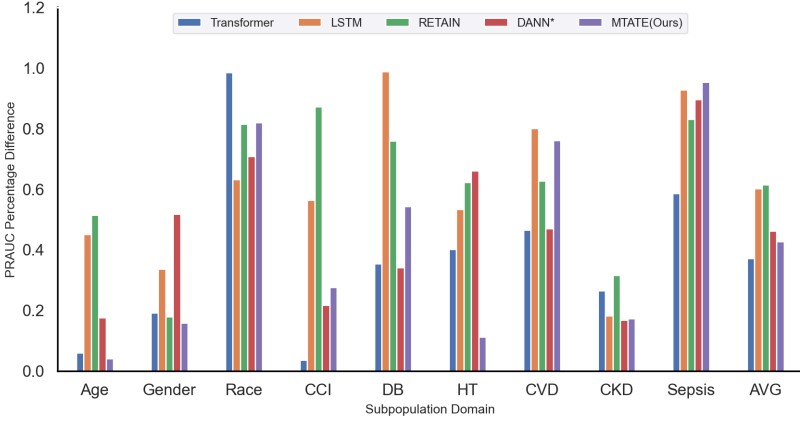

Figure A2: The comparison between MTATE with baseline methods for the percentage difference score in PRAUC for each domain. Y-axis represents the percentage difference. X-axis represents the subpopulation domain, each domain consists of two subpopulations (e.g., Young ($< 65$ y/o) vs. Old in Age domain, Sepsis vs. Non-Sepsis in Sepsis Domain ). CCI stands for charlson comorbidity score, DB stands for diabetes, HT stands for hypertension, CVD stands for cardiovascular disease, CKD stands for chronic kidney disease.

## A.4 STUDY OF MASKING RATE

We analysed the effect on the performance by different masking ratio. Figure A3 shows the performance metrics of ROCAUC, PRAUC and accuracy with respect to the masking ratio from 0 to 0.9. Both ROCAUC and PRAUC have a similar trend, both starting with a relative high score and gradually decrease with some punctuation, and both reach lowest scores at masking rate 0.8 and 0.9. In contrast, the accuracy has a opposite trend, where it starts with the lowest score, and almost monotonically increasing. The masking rate is highly dependant on the input data, thus the chosen of masking rate is not universal. For our experimental data, one of the best masking rate is 0.4, where it has the highest accuracy and PRAUC and the third best ROCAUC.

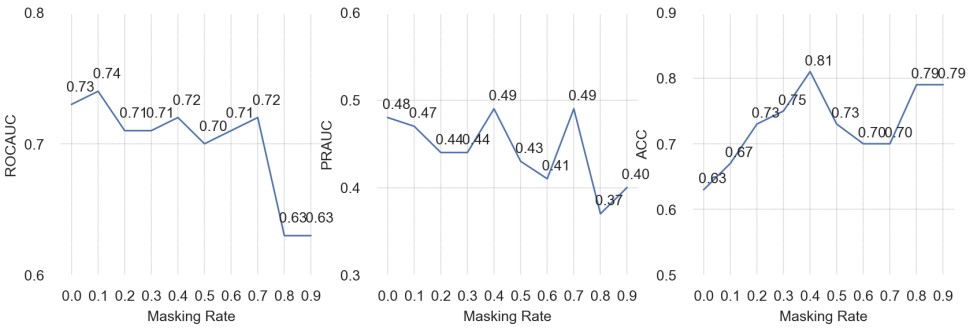

Figure A3: Performance Score of ROCAUC, PRAUC and Accuracy with different masking rate

## A.5 RELATIONSHIP BETWEEN OUTCOME LOSS, DOMAIN LOSS AND REPRESENTATION-WISE ATTENTION

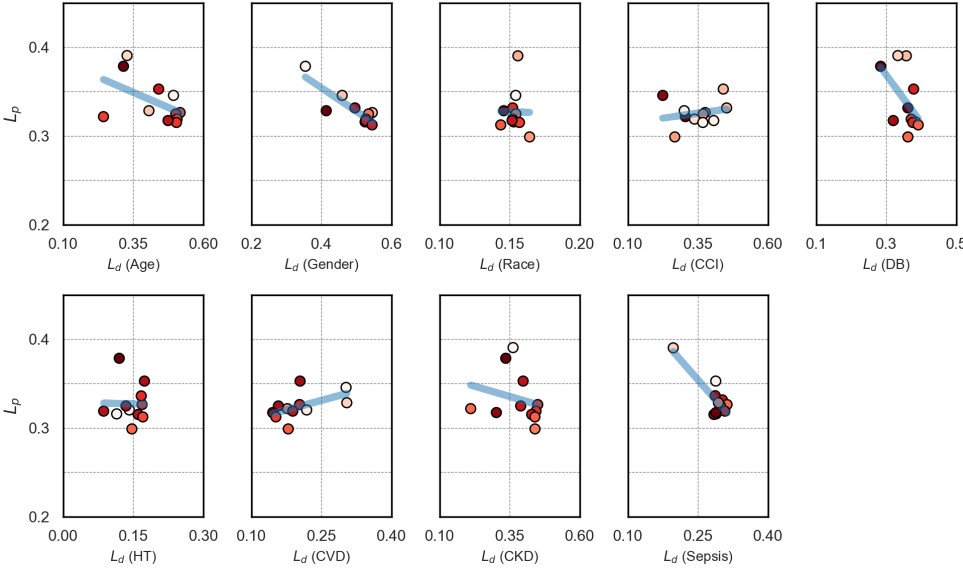

Figure A4: Relationship between outcome loss, domain loss and representation-wise attention in all domains. Y-axis represents the outcome loss, x-axis represents the domain loss. The colored dots represent the representation-wise attention, and darker color represents higher attention.

