# OpenReview forum: "Unbiased Representation of Electronic Health Records for Patient Outcome Prediction"
_ICLR.cc/2023/Conference — Submitted to ICLR 2023_

### Official Review · Reviewer_M4Hz · 2022-10-24

**Confidence:** 3
**Correctness:** 3
**Technical Novelty And Significance:** 3
**Empirical Novelty And Significance:** 2
**Recommendation:** 3

**Clarity, Quality, Novelty And Reproducibility:**


Clarity
- this paper is very clear

Quality
- the methods and experiments all seem sound

Novelty
- I think there are issues with the novelty of this work (see "weaknesses" above)

Reproducibility
- The authors don't mention that they will share code, but their explanations of their methods are fairly clear, so I think someone could independently reproduce their method.



**Strength And Weaknesses:**


Strengths:
- This paper is well written
- it focuses on an important problem (they also motivate the problem very well)
- the methods seem sound, though I am not an expert
- the experiments use relevant data, and the results demonstrate that their method works well
- their experiments use public data

Weaknesses
- There has been a lot of work on this topic, and the authors don't cite or compare with existing methods. Here is a short list of papers that look relevant to this topic, but were not mentioned or compared with in the paper:
-- https://doi.org/10.1109/ICDM50108.2020.00050
-- https://doi.org/10.1145/3394486.3403129
-- https://doi.org/10.1109/ICDM51629.2021.00060
-- here is a relevant review: https://doi.org/10.1016/j.jbi.2020.103671
- there is no comparison against state-of-the-art risk prediction models in medicine, or any non-DL models at all. there are several such models used in medicine (which don't use DL), which would be good baselines to compare against here.
- the authors don't provide any code



**Summary Of The Paper:**

The authors present a transformer-encoder framework for multipleThe authors present a encoder/transformer method for learning representations of EHR data,  for multiple prediction tasks. They demonstrate that their method out-performs other DL methods on one prediction task using MIMIC-III data.


**Summary Of The Review:**

This is a well-written paper on an interesting and important topic. The experiments are somewhat compelling andt he methods appear to be sound. The main issue is that the authors don't disucss or compare with other methods from the literature. There has been a lot of work in the encoder/transformer space for EHR data, and the authors don't engage with this literature. Since they are presenting their method as a general approach for EHR data (and not a method specific to AKI), they should make sure to engage & compare with exisitng work.

---

> ### Author Response · Authors · 2022-11-18
> **Responses to Reviewer M4Hz**
>
> We thank the reviewer for your valuable comments and suggested literature. Given the limited time, we will not be able to add comparisons to more baselines. We will add more baseline methods for the comparison in future work. We will share the GitHub link of the code upon the acceptance of the paper.

---

### Official Review · Reviewer_H5nZ · 2022-10-25

**Confidence:** 4
**Correctness:** 3
**Technical Novelty And Significance:** 2
**Empirical Novelty And Significance:** 3
**Recommendation:** 5

**Clarity, Quality, Novelty And Reproducibility:**

Clarity:
The paper is well written, organized, and technically detailed to a satisfactory extent. The notation is clear and consistent throughout the paper.

Quality:
The design and justifications for the proposed Masked Triple Attention Transformer Encoder are technically sound. The observations made regarding the improvements in patient outcome prediction performance introduced by the proposed MTATE are empirically well supported. Overall, the work seems to be fairly well developed in terms of technical quality.

Novelty:
From a methodological perspective, the contribution of this work can be considered rather incremental. In essence, the authors leveraged several existing attention-based modules and integrated them into a single architecture for the purposes of learning EHR representations that preserve different types of associations within the data. Nevertheless, each of these attention mechanisms are already established in the literature which limits the novelty of this work.

Reproducibility:
The reproducibility of this work is somewhat limited since all experiments have been conducted on, what appears to be, a proprietary dataset which has not been made publicly available by the authors. Even if the authors could not make this dataset available, they should have evaluated MTATE and the baselines on at least one widely-used publicly available EHR dataset (e.g., MIMIC-III [1]). As for the code for the proposed MTATE model, it is also not made available by the authors, however, the architecture of the model is clearly explained in the paper, thus it is safe to say that one should be able to implement MTATE by following its description in Section 2.

[1] Johnson, A. E., Pollard, T. J., Shen, L., Lehman, L. W. H., Feng, M., Ghassemi, M., ... & Mark, R. G. (2016). MIMIC-III, a freely accessible critical care database. Scientific data, 3(1), 1-9.


**Details Of Ethics Concerns:**

Not applicable.

**Strength And Weaknesses:**

Strengths:

* The problem addressed in this paper is of considerable importance as approaches that account for patients’ backgrounds and subpopulations are quite relevant to personalized medicine and reliable prediction of patient outcomes.

* The proposed MTATE model is capable of learning optimal yet fair representations of EHR sequences, which is of great importance to outcome prediction for patients with different demographic characteristics and clinical histories.

* MTATE incorporates time-wise, feature-wise and representation-wise attention mechanisms. This allows for (1) capturing associations among timesteps (which are quite informative in the case of EHR data), associations among features, and (2) summarizing feature-attended representations for downstream patient outcome prediction.

* While it is unclear whether domain-invariant or domain-specific data representations should be used for prediction tasks concerning EHR data, the authors claim that MTATE learns EHR representations that can be domain-specific, domain-invariant, or even a mixture of the two.

* Experiments conducted on a real-world EHR dataset demonstrate that MTATE outperforms several outcome prediction baselines on mortality risk prediction for critically ill AKI-D patients, yielding improvements in both classification metrics as well as fairness metrics, thus mitigating bias towards different subpopulations in the data.

* A few traditional classifiers (XGboost, SVM and RF) have achieved performance comparable to that of MTATE and outperformed some of the compared baseline methods; which suggests that MTATE’s representations can also serve as pre-trained EHR data representations for various classifiers applied to downstream prediction tasks.

* The authors conducted an insightful analysis of the relationship between the patient outcome loss, the domain loss and the representation-wise attention.

-------------------------------------------------------------------------------------------------

Weaknesses:

* The proposed MTATE model appears to resemble a multi-task attention classifier with an output layer that combines the representations learned for all ‘tasks’ (i.e. domains) into a single outcome. Due to this resemblance, I believe that a comparison of MTATE to multi-task classification models should have been carried out. I would encourage the authors to consider such a comparison or provide their reasoning as to why multi-task classification models should not be considered among the baselines.

* In Section 2.3, the authors state the following: “Since features for the different domains are not equally important, we randomly masked out some number of latent features along all time steps differently for each FR-Attention module. In return, the masking procedure forces each sub-module to learn different feature focuses for each domain to generate unique domain representations.” However, no experiment has been conducted to inspect the impact of the random masking on the learned feature representations and thus on the overall patient outcome prediction performance. I am aware of the ablation study (Table 3) conducted to assess the performance with and without masking, nevertheless, such a study does not provide an insight into the effect of different random masking trials. I would suggest that the authors elaborate on this aspect of the random marking in their response.

* One of my main concerns is that the resulting representations learned for each domain can be domain-specific or domain-invariant, depending on the domain classification loss. Thus, target domains that are “easier to classify” will tend to have smaller loss values. On the other hand, other target domains might be difficult-to-classify yet contain more informative patterns, in which case such domains will have higher loss values and may need many iterations for their losses to be substantially minimized. This case does not appear to be accounted for in the model design. I would encourage the authors to clarify the potential behavior of MTATE in the aforementioned scenario.

* In Section 2.6, the authors mention that “the final representation C of EHR is concatenated with all static features, such as demographics and comorbidity, followed by a linear layer with a sigmoid function for the outcome binary prediction”. I assume that the demographics and comorbidity features are one-hot encoded, in which case it would be desirable that the range of the learned representations’ values is around [0, 1]. Is a certain normalization technique applied to the final representations to scale them within the same (or similar) range as that of the static features? If this is not the case, I would like to ask the authors to provide their thoughts on this point in their response.

* The authors have generated 30 samples from each patient’s EHR history with a random start and end times. Nevertheless, I am wondering whether there is a specific reason behind this data preprocessing choice. Why not consider the entire EHR sequence for each patient? Were the sequence samples generated just for data augmentation purposes, i.e. so that representations can be learned from a larger dataset created by generating multiple samples from a single EHR sequence?

* In Section 3.2, the authors mention that all models have been compared on both imbalanced and balanced sets having positive/negative ratios of 1:4 and 1:1, respectively. However, I am not certain if 1:4 can be considered imbalanced (of course, the ratio itself is not a balanced ratio, but the classification problem cannot be considered a typical case of an imbalanced classification problem, or at least not a highly imbalanced classification problem).

* In addition to the considered baselines, I believe that MTATE should have been compared to several recently published state-of-the-art representation learning methods for EHR data including, but not limited to, the methods proposed in:

    Darabi, S., Kachuee, M., Fazeli, S., & Sarrafzadeh, M. (2020). Taper: Time-aware patient ehr representation. IEEE journal of biomedical and health informatics, 24(11), 3268-3275.

    Bang, S. J., Wang, Y., & Yang, Y. (2020). Phased-lstm based predictive model for longitudinal ehr data with missing values.

    Bai, T., Zhang, S., Egleston, B. L., & Vucetic, S. (2018, July). Interpretable representation learning for healthcare via capturing disease progression through time. In Proceedings of the 24th ACM SIGKDD international conference on knowledge discovery & data mining (pp. 43-51).

-------------------------------------------------------------------------------------------------

Minor weaknesses:
There are also certain grammatical and typographical errors and remarks that require attention. Some of them are summarized as follows:
- First paragraph on page 4: The sentence “Thus, the attention weights computed from the query and key represent how much focus each time step is associated with other time steps.” is rather unclear and should be revised accordingly.
- Page 4: In the last sentence of Section 2.4, “generate” should be replaced with “generating” and a word seems to be missing between “for” and “in”.
- Page 5: “C” should be bolded in the paragraph after Eq. (9).
- Last paragraph on page 6: In the sentence “LSTM is the most competitive method since it has the same highest accuracy”, I assume that the authors meant “same highest ROCAUC” instead of “same highest accuracy”.
- Page 7, last paragraph of Section 4.1: I believe that “Figure 4.1” should be replaced with “Figure 3”. Later in the same paragraph, “Figure A.3.2” should be replaced with “Figure A4”.
- In Section 4.3: replace “negative correlated” with “negatively correlated”.


**Summary Of The Paper:**

This paper addresses the fairness problem in a healthcare setting involving outcome prediction from electronic health records (EHR). To this end, the authors propose the Masked Triple Attention Transformer Encoder (MTATE), an adaptive multi-task learning model capable of automatically learning and selecting optimal yet fair data representations without explicit domain adaptation or domain-specific bias correction. The experimental results suggest that MTATE learns representations of diverse subpopulations that lead to improvements in predictive performance and fairness on the task of mortality risk prediction for critically ill Acute Kidney Injury requiring Dialysis (AKI-D) patients.

**Summary Of The Review:**

Overall, this paper proposes an adaptive multi-task learning model capable of learning optimal and fair data representations without explicit domain adaptation or domain-specific bias correction. Addressing fairness in a healthcare setting involving patient outcome prediction is of considerable importance. Nevertheless, the work lacks methodological novelty and is rather incremental in that respect considering that it combines already established attention mechanisms. That being said, I recommend that this work is not considered for acceptance at ICLR due to reasons including, but not limited to, the following:

(1) the degree to which this work is application-oriented rather than a work that makes novel methodological contributions in representation learning;

(2) the absence of a comparison of the proposed model to more recently published models for representation learning from EHR data;

(3) unjustified data preprocessing decisions (outlined in the list of weaknesses);

(4) experiments conducted solely on proprietary data despite the availability of public EHR datasets that also contain records of AKI-D
patient among other conditions.

Despite the above reasons, I am still looking forward to the authors’ response and I would be willing to adjust my score in case I have misunderstood or misinterpreted certain aspects of the work.

---

> ### Author Response · Authors · 2022-11-18
> **Responses to Reviewer H5nZ**
>
> We thank the reviewer for your time and insightful feedback that will help us improve the manuscript. Please see below our responses to your questions.
>
> **The scenario of difficult-to-classify domains contain informative patterns**:
> Thanks for the great question.  One solution to this situation is to increase the number of iterations so that all the domain losses are minimized to some extent.  Nonetheless, if we stop the iteration before they all get minimized (when some domains have higher losses, and some have lower losses), the learned representation with a higher loss can be regarded as the domain-invariant features, the one with lower loss can be regarded as the domain-specific features, and both features could contain informative patterns. Thus, our proposed model does not lean toward any one of them, we used another type of attention (i.e., representation-wise attention) to select the optimal one for the downstream prediction task.  Also, the domain loss that is used for representation-wise attention is normalized across domains. Thus, it is relative domain loss where there is always a higher or lower loss. Again, we let the model decide which representation is more appropriate for the prediction task.
>
> **The scale of final representations C**:
> Yes. The demographic and comorbidity features are one-hot encoded. We used softmax normalization along the feature dimension in the representations to ensure they are within the same range as the static features.
>
> **Sample generations**:
> We generated 30 samples from each patient's EHR history for two purposes. First, and yes, one of the purposes is for augmentation. Secondly, one of the major research questions for our experiment is to continuously predict the patient' outcome given the patient data at any period during the hospitalization. Thus, for each new patient, we can predict the outcome over time given the data within some duration so that a better monitoring process can be given.
>
> **Grammatical and typographical errors**:
> We have updated our paper regarding those errors. Moreover, we rephrased the unclear sentence with this new one in the paper: "Thus, the attention weights computed from the query and key represent how much focus all features at one single time steps is associated with themselves at other time steps."
>
> **The impact of random masking**:
> Thanks for pointing this out!  We will design and conduct more experiments to investigate the impact of random masking in future work. For example, if I understood it correctly, we can add the comparison between the random masking and the masking on the same feature for every domain. I would appreciate it if you could tell me a little more about this concern.
>
> **Multi-task models, other baselines, and public dataset**:
> Thanks for raising the topic of multi-task models shared with us lots of representation learning models. We have reviewed some relevant models that use the multi-task learning framework to address the fairness issue, as well as other representation models for fairness. Nonetheless,  given the limited time, we will not be able to add comparisons to the more baselines. In future work, we will add more baseline methods and evaluate the proposed model on the public EHR dataset.

---

> > ### Comment · Reviewer_H5nZ · 2022-11-20
> > **Follow-up on authors' response**
> >
> > I would like to thank the authors for their point-by-point response to my review. The following summarizes my thoughts on some of the major points from the response:
> >
> > **Sample generations:** Considering your latter point on continuous prediction of a patient outcome given the patient data at any period during the hospitalization, I am wondering why windowing was not applied to extract all consecutive subsequences (of fixed or even variable length) from each patient’s EHR history instead of randomly sampling 30 subsequences? In fact, a windowning approach would have (1) achieved the augmentation purpose (indeed, it would have generated even more samples than the random sampling approach) and (2) covered more (sub-)periods during a patient’s hospitalization.
> >
> > **The impact of random masking:** I do agree with the authors that a comparison between random masking and masking of the same feature(s) for every domain would also be a relevant experiment. However, what I was referring to in my initial review was an experiment in which different repetitions (trials) of random masking are applied and the downstream performance is measured after each repetition. More specifically, the authors can generate one random masking of the features in the different domains and subsequently measure the downstream outcome prediction performance; then, a different random masking can be generated and applied to the features after which the downstream performance should be also measured. The authors can repeat this procedure multiple times (e.g., 30, 50 or even 100 times), and calculate the mean and standard deviation of the classification/fairness metric values that were measured across all repetitions of the experiment. These aggregate metrics can provide an insight into how “sensitive” or “robust” the MTATE model is with respect to the random masking procedure, i.e. how much does the random masking procedure affect MTATE’s outcome prediction performance as well as its fairness.
> >
> > **Multi-task models, other baselines, and public dataset:** I do understand that, given the limited time frame, the authors will not be able to consider the additional baselines that I suggested and evaluate all modes on a public EHR dataset. Nevertheless, I hope that the authors will consider these suggestions in their future work.

---

### Official Review · Reviewer_UWzS · 2022-10-25

**Confidence:** 3
**Correctness:** 3
**Technical Novelty And Significance:** 2
**Empirical Novelty And Significance:** 3
**Recommendation:** 6

**Clarity, Quality, Novelty And Reproducibility:**

The method was presented clearly with formulas, and the experiments were well conducted. It connects domain-specific and domain-invariant representations to achieve fairness and better model prediction performance.

However, the code was not provided for reproducibility purposes. Also, are there any tuning parameters for the proposed method? How were they selected and how much do they affect the model results?


**Strength And Weaknesses:**

The experiment was well conducted. The proposed model has shown advantages in both prediction performance and fairness across prespecified subgroups.

More explanation and intuitions can be provided for the proposed method, for example, the rationale for introducing each model part and their connection with existing methods. Also, the reason why the model can achieve desired fairness can be better described.



**Summary Of The Paper:**

In this paper, the authors propose an attention-based encoder model to learn data representations that are fair for prespecified subpopulations. The proposed method was tested on real data to learn EHR representations and predict 72-hour mortality after dialysis.

**Summary Of The Review:**

The method in the paper is proposed to balance domain-specific and domain-invariant representations with the aim of learning fair and unbiased representation. The experiment has shown its advantage empirically. Overall, I think the proposed method could be useful. But more details can be provided in the paper.

---

> ### Author Response · Authors · 2022-11-18
> **Responses to Reviewer UWzS**
>
> We thank the reviewer for your time and your valuable comments. Please see our responses below:
>
> **Intuitions**:
> Our proposed model's intuition is to integrate domain-specific and domain-invariant features in one model. One of the most innovative parts is using the domain classifier loss to learn the attention to get a weighted representation for the downstream task. The process of learning fair representations and optimization of the downstream prediction results are trained simultaneously, which is another improvement of the existing methods where most of the fair representation models in the literature require learning the representation and downstream task in separate steps. Also, they focus on either domain-specific or domain-invariant features, never on both.
>
> **Hyperparameters and reproducibility**:
> We chose the hyperparameters that delivered the best performance for the validation set by doing random searches for all methods. The resulting hyperparameters for MTATE  are:  The number of encoder layers to stack is 1. The dimensions of the representation-wise attention weights are 8 and 1. The dimension for temporal relevance attention is equal to the number of features, and the feature relevance attention is equal to the number of time steps. The learning rate is 0.01, the batch size is 512, and the dropout rate is 0.01.  We will share the GitHub link of the code upon the acceptance of the paper.

---

### Official Review · Reviewer_icM1 · 2022-10-31

**Confidence:** 3
**Clarity, Quality, Novelty And Reproducibility:** Not easy to follow. Method novelty is…
**Correctness:** 3
**Technical Novelty And Significance:** 2
**Empirical Novelty And Significance:** 2
**Recommendation:** 3

**Strength And Weaknesses:**

Strength:
- The problem is very important especially in health care domain.

Weakness:
- The paper is not easy to follow
- I think using the word "domain" might by non-intuitive in this context. It is usually referred to as sensitive or protected groups.
- The method seems to be adhoc without a principle way to motivate the structure of the network.
- It is claimed in the paper that "To the best of our knowledge, MTATE is the first model that automatically trains and determines optimal and fair data representations." There are many papers about learning fair representation. The one that needs to be highlighted here is "Creager, Elliot, et al. "Flexibly fair representation learning by disentanglement." International conference on machine learning. PMLR, 2019." The method in that paper learns representation that is invariant to the sensitive groups, which is the main objective of the current paper.
- Missing important baselines to compare to.

**Summary Of The Paper:**

The authors propose a method to learn representation that is invariant to sensitive groups (or domains as claimed by the authors). This is achieved through a network structure that has a transformer to generate time-wise attention, domain-specific feature-wise attention (to extract features that are specific to each domain/subgroup), and finally all these representations are concatenated and passed to a network to generate a universal representation that can be used for downstream tasks.

**Summary Of The Review:**

Read the strength and weakness section.

---

> ### Author Response · Authors · 2022-11-18
> **Responses to Reviewer icM1**
>
> We thank the reviewer for your thoughtful comments and suggestions. We have updated our paper for a clearer representation.
>
> **The “domain”**: We agree that the word “domain” is a bit non-intuitive in this specific experiment which focuses on the subpopulations or protected groups. However, we chose the “domain” because the proposed framework is not limited to subpopulation applications. It can also be used in more general cases, where it can be used for representation learning for multiple data sources (e.g., data collected from different hospitals), which are considered as different domains.
>
> **Innovation**:   We agree that the claim “MTATE is the first model that automatically trains and determines optimal and fair data representations.” is ambiguous. However, we wanted to mean that MTATE is the first model that integrates both domain-specific and domain-invariant features in one model. It trains the fair representation and predicts the downstream task altogether. While most of the fair representation models in the literature require learning the representation and downstream task in separate steps, also they focus on either domain-specific or domain-invariant features, never on both.
>
> **Baselines**: Thank you for suggesting the excellent related work! Given the limited time, we will not be able to add comparisons to more baselines. We will address these issues in future work. We will share the GitHub link of the code upon the acceptance of the paper.

---

### Decision · Program_Chairs · 2023-01-20

**Decision:**

Reject

**Justification For Why Not Higher Score:**

There is still insufficient enthusiasm from the referees to accept this paper after rebuttals.

**Justification For Why Not Lower Score:**

N/A

**Metareview: Summary, Strengths And Weaknesses:**

This paper studied how to learn unbiased and fair representations of different subpopulations from their electrical health records (EHR). The referees all agree the paper addressed an important and interesting problem. However, they also raised a series of concerns, including insufficient comparisons with the methods in the literature, and insufficient explanation or intuitions about the procedure. There is still insufficient enthusiasm from the referees to accept this paper after rebuttals.